# Profiles of Motor-Cognitive Interference in Parkinson’s Disease—The Trail-Walking-Test to Discriminate between Motor Phenotypes

**DOI:** 10.3390/brainsci12091217

**Published:** 2022-09-09

**Authors:** Thomas J. Klotzbier, Nadja Schott, Quincy J. Almeida

**Affiliations:** 1Institute of Sport and Movement Science, University of Stuttgart, 70569 Stuttgart, Germany; 2Movement Disorders Research & Rehabilitation Centre, Department of Kinesiology, Wilfrid Laurier University, Waterloo, ON N2L 3C5, Canada

**Keywords:** dual-task, Trai-Walking Test, gait disorder, diagnosis, motor-cognitive interference, Parkinson’s disease

## Abstract

Background and Aims. Most research on Parkinson’s disease (PD) focuses on describing symptoms and movement characteristics. Studies rarely focus on the early detection of PD and the search for suitable markers of a prodromal stage. Early detection is important, so treatments that may potentially change the course of the disease can be attempted early on. While gait disturbances are less pronounced in the early stages of the disease, the prevalence, and severity increase with disease progression. Therefore, postural instability and gait difficulties could be identified as sensitive biomarkers. The aim was to evaluate the discriminatory power of the Trail-Walking Test (TWT; Schott, 2015) as a potential diagnostic instrument to improve the predictive power of the clinical evaluation concerning the severity of the disease and record the different aspects of walking. Methods. A total of 20 older healthy (M = 72.4 years, SD = 5.53) adults and 43 older adults with PD and the motor phenotypes postural instability/gait difficulty (PIGD; M = 69.7 years, SD = 8.68) and tremor dominant (TD; M = 68.2 years, SD = 8.94) participated in the study. The participants performed a motor-cognitive dual task (DT) of increasing cognitive difficulty in which they had to walk a given path (condition 1), walk to numbers in ascending order (condition 2), and walk to numbers and letters alternately and in ascending order (condition 3). Results. With an increase in the cognitive load, the time to complete the tasks (seconds) became longer in all groups, *F*(1.23, 73.5) = 121, *p* < 0.001, *ɳ*^2^_p_ = 0.670. PIGD showed the longest times in all conditions of the TWT, *F*(2, 60) = 8.15, *p* < 0.001, *ɳ*^2^_p_ = 0.214. Mutual interferences in the cognitive and motor domain can be observed. However, clear group-specific patterns cannot be identified. A differentiation between the motor phenotypes of PD is especially feasible with the purely motor condition (TWT-M; AUC = 0.685, *p* = 0.44). Conclusions. PD patients with PIGD must be identified by valid, well-evaluated clinical tests that allow for a precise assessment of the disease’s individual fall risk, the severity of the disease, and the prognosis of progression. The TWT covers various aspects of mobility, examines the relationship between cognitive functions and walking, and enables differentiation of the motor phenotypes of PD.

## 1. Introduction

In addition to the motor symptoms, various aspects of cognitive impairment in Parkinson’s disease (PD) patients can have a negative impact on the ability to balance in static and dynamic situations [1,2]. The extent of the cognitive impairment is heterogeneous in those affected and worsens in the course of the disease parallel to the motor symptoms [3]. The prevalence of comorbid dementia is estimated at 26–44% [4,5], with main deficits being found in the executive/attentional, memory, and visuospatial domains [4], which can magnify their gait problems. In their multicenter study, Hely and colleagues [6] reported that at least 83% of survivors had dementia after 20 years. In particular, walking with additional motor or cognitive tasks to be performed in parallel seems difficult for those affected [7,8,9]. This is particularly significant because walking in the real world—usually under dual task (DT) conditions—requires attention to various changing environmental features to avoid tripping and slipping and to recover quickly from unavoidable postural disturbances. Therefore, it is not surprising that deficits in attention and executive functioning (EF) are independently associated with risk for postural instability, impairments in activities of daily living (ADL), and future falls [10,11,12]. Individuals with PD who have cognitive decline appear to be more susceptible to gait impairments due to their inability to use cognitive resources required to plan and control movements, especially when the automaticity of well-learned movements (gait) is compromised and where increased conscious control is required [13].

Although the diversity of DTs found in the literature makes a comparison between the studies difficult, the gait of individuals with PD is more influenced during the performance of more complex secondary tasks [14]. A meta-analysis by Ruffegeau and colleagues [15] demonstrated sufficient evidence to conclude that DT conditions involving EF skills significantly hinder walking in people with PD, despite variation between study paradigms. Moreover, DT paradigms with additional cognitive tasks can be helpful to parse apart the tremor dominant (TD) patients that will progress slower than those that are assigned to the phenotype postural instability/gait difficulty (PIGD) and will progress faster [16]. However, there is some controversy in this regard. While some studies report that the TD phenotype has a better prognosis and a lower rate of disease progression compared to the PIGD phenotype, others claim that there is no difference in long-term outcomes [17,18,19,20].

While most studies with the DT paradigm use relatively simple straight walking as a locomotion task—and also the guidelines of the Canadian Consortium on Neurodegeneration in Aging (CCNA) only address recommendations for straight walking [21]—complex locomotion tasks in which the walking speed is adjusted and the walking direction change seem to be particularly demanding and sensitive for PD patients to produce dual-task costs (DTC). While simple information processing processes can solve straightforward walking, cognitive flexibility and the ability to change tasks explain the speed of cornering [22] and walking with direction changes [23]. During complex walking situations (walking with direction changes), the increased cognitive and sensory processing required to plan gait modifications may strongly impact the walking performance [24]. Difficulties in turning around the body axis are one of the most common complaints among people with PD, may cause extreme gait slowness and loss of balance and may result from an overloaded or inefficient cognitive system in PD when planning complex gait adjustments. For this reason, and since walking performance is most affected by internal disturbances [25,26], a mobile version of the Trail-Making Test (TMT) is used in this study (Trail-Walking Test; [27]) as a motor-cognitive DT that demands EF with varying degrees of difficulty.

Research usually focuses on the early detection of PD and the search for suitable markers of a prodromal stage [28]. So far, there is no gold standard for the operationalization of gait disorders in PD. The walking test (Item 29) integrated into the motor part of the Unified Parkinson’s Disease Rating Scale (UPDRS; Movement Disorder Society Task Force on Rating Scales for Parkinson’s Disease, [29]) is frequently used in clinical settings. However, the PIGD score does not include the classification of freezing of gait, does not capture the performance of tandem gait, and lacks details about the range of postural deficits [30]. In order to improve the predictive power of the clinical examination concerning the risk of falling or to differentiate between the PD motor phenotypes, it appears necessary to record not only reactive and supportive aspects of balance control but also anticipatory, arbitrary, and cognitive aspects of locomotion.

The aim was to compare single task (ST) and DT conditions concerning a possible detection of PIGD and whether it is possible to differentiate the groups based on the TWT, what might be important for the timing and duration of therapy initiation and may help to assist with the prognosis and the tailoring of treatment. Based on the difficulties in the mentioned motor and cognitive domains in PD patients, we hypothesized that overall, individuals with PIGD perform more poorly than the control group and the TD group in all conditions of the TWT. We also assume that individuals with PIGD exhibit proportionally greater DTC under more complex, attention-demanding motor-cognitive DTs (TWT conditions) relative to TD patients or healthy older adults (see also [31] in people with mild cognitive impairment, MCI).

## 2. Materials and Methods

### 2.1. Participants

Table 1 shows the inclusion and exclusion criteria of the control group, PD patients with TD, and PD patients with PIGD. A total of 20 healthy older adults and 43 older adults with PD voluntarily participated in the study. Based on the Unified Parkinson Disease Rating Scale—Part 3; UPDRS; [32]—two motor phenotypes of PD patients were distinguished: the tremor dominant (TD) and the postural instability (PIGD) motor phenotype (classification according to Jankovic et al. [33]; see Table 2). Participants were invited in writing or orally (by telephone) to the Sun Life Financial Research and Rehabilitation Centre for Movement Disorders (MDRC) at Wilfrid Laurier University in Waterloo, Canada. The subjects with PD were asked to postpone their medication intake by 12 h before visiting the clinic to participate in the study without medication (“off-state”) to minimize the confounding effect of dopaminergic medication on cognitive and motor performance, especially gait speed [34,35]. 

### 2.2. Instruments

#### 2.2.1. Sociodemographic Information, Cognitive Performance, and Fall-Associated Self-Efficacy

Demographic information, medical history, physical activity, and the number of falls in the last year were collected using questionnaires. In addition, the height and weight of the participants were measured, and the body mass index (BMI, kg/m^2^) was calculated.

Since cognitive status influences strategies for allocating attentional resources [36,37] and Johansson et al. [38] have shown in a recently published study that PD patients with MCI use a posture-first strategy and had larger DTCs on gait than PD non-MCI patients, it is important to consider the cognitive status. Although it is recommended to use a comprehensive cognitive assessment battery rather than individual global cognitive measures to assess the cognitive state, we used the well-established Montreal Cognitive Assessment (MoCA; score range: 0–30; [39]) to test general cognitive performance. This instrument appears to be sensitive to slight cognitive loss (mild cognitive impairment; MCI) in cognitively intact older adults [40,41].

The paper-pencil-based Trail-Making Test [42] comprises 25 circles to be connected (∅13 mm), which are numbers (Part A; visuomotor skills, visual processing speed) and numbers or letters (Part B; working memory, cognitive flexibility, executive functions, and visuo-spatial skills). The aim is to connect the numbers in ascending order from 1 to 25 (Part A) and the numbers in ascending order from 1 to 13 alternately with the letters in alphabetical order from A to L (Part B) in the shortest possible time without error. In addition, we introduced a motor speed condition with the task of following a predefined path connecting 25 circles [43] with (a) the idea of calculating the purely cognitive performance of the task of connecting numbers or numbers and letters without the influence of motor performance (moving the stylus, which can be difficult, especially for people with TD) and (b) to calculate the cognitive DTC.

The Activities Specific Balance Confidence (ABC) Scale [44] assessed fall-associated self-efficacy. On a scale of 0–100%, the participants should estimate their confidence to carry out 16 activities without becoming unbalanced. High percentages stand for a high fall-associated self-efficacy.

The Timed Up and Go (TUG; [45]) test is one of the most common tests used to examine balance, gait speed, and functional ability related to the performance of basic activities of daily living (ADL) in older populations. The TUG measures the time (seconds) it takes a participant to stand up from a chair, walk 3 m at a comfortable speed, walk around a cone, walk back, and sit down on the chair. With a “cut-off” value of 14 s or more, the TUG is considered a good predictor for identifying healthy individuals at risk of falling [46,47].

The Geriatric Depression Scale (GDS) according to Sheikh and Yesavage [48] comprises in short form 15 questions. The GDS allows for early detection of possible depression in aging patients. Scores between 0–5 are normal, scores between 6–10 indicate mild to moderate depression, and scores between 11–15 indicate severe depression.

#### 2.2.2. Rating Scale for Parkinson’s Disease

The UPDRS [32] is divided into the areas of (1) cognitive functions, behavior, and mood, (2) activities of daily living (ADL), (3) motor examination, and (4) complications of treatment. The motor dimension of UPDRS was used to determine PD cardinal symptoms of tremor, rigor, bradykinesia, and postural instability. Based on the evaluation and the classification algorithm according to Jankovic et al. [33], this scale allows for a classification of the mentioned PD motor phenotypes: tremor dominant type (TD = mean value of points for tremor/mean value of points for PIGD ≤1.5) and motor phenotype with postural instability and gait disorder (PIGD = mean value of points for tremor/mean value of points for PIGD ≥1.0). The following items were used to evaluate the two motor phenotypes:

#### 2.2.3. Trail-Walking Test

The TWT [27] was used to assess motor-cognitive interference under change in direction walk conditions. In this motor-cognitive DT, 15 hats with banderoles are placed randomly in a 16 m^2^ area (4 × 4 m). A 30 cm (diameter) circle was drawn around each hat. The TWT has three different conditions. In the first condition (TWT-M, ST), participants were instructed to follow a line connecting 15 circles (purely motor task). In condition 2 (TWT-A, DT), participants were asked to navigate to numbered targets in sequential and ascending order from 1 to 8. In a third and more cognitive demanding DT (TWT-B), participants were instructed to step on targets with an ascending, alternating number(1-8)-letter(A-G) sequence (1-A-2-B-3...-8) (see Appendix A: Appendix A). Participants were asked to perform the task as accurately but as quickly as possible in all conditions. No priority was given to one domain or the other. Time per trial was taken with a stopwatch to the nearest 0.01 s, and motor errors (e.g., knocking over a hat or not stepping inside the circle) and shifting and sequencing errors (e.g., navigating to the wrong number/letter; [48]) were recorded. Sequence and shift errors were corrected promptly by the test administrator asking the participant to return to the last correct circle. Errors are reported and accounted for in the required times since the correction of errors takes extra time [43]. Each condition was performed three times.

### 2.3. Procedure

The participants were informed about the objectives and contents of the planned study and the test procedure, test duration, and possible risks of data collection. Before the data collection was carried out, a written declaration of consent was obtained. The methods used in these studies are in accordance with the ethical principles of the Helsinki Declaration [49], national legislation and relevant international norms and standards. The implementation of the procedures was randomized to avoid possible sequence effects. All tests were performed in a quiet environment to avoid distractions and to exclude possible interfering variables in the experimental situation. The majority of the participants could be tested within the planned 90 min. In order to keep the effect of fatigue to a minimum, a rest time of 1–3 min between the conditions and trials was made. None of the participants experienced complications during data collection. Previously trained test administrators carried out the data collection. The research project received ethical approval (REB # 4791 Project, “Motor-cognitive interference in dual tasks: allocation of resources in Parkinson’s Disease patients” REB Clearance Issued: 19 February 2016). Participants were recruited in March 2016. Data collection also took place in March 2016.

### 2.4. Data Analysis

All statistics were performed with SPSS v.27 (SPSS, Chicago, IL, USA). We first explored dependent variables to examine missing values, normality of distributions (Kolmogorov–Smirnov tests), and presence of outliers (defined by the Explore command of SPSS v.27).

For the sample description, between group differences for continuous variables (e.g., age, height, weight, BMI, physical activity) were calculated using ANOVAs; partial eta^2^ was calculated as an effect strength measure (Conventions of Cohen, [50]: 0.01 small effect; 0.06 medium effect; 0.14 strong effect). In addition, categorical demographic variables were tested with a Chi^2^ test (e.g., sex).

To test the effect of different cognitive conditions and difficulty levels, a 3 (group: Control, TD, PIGD) × 3 (condition: TWT-M, TWT-A, TWT-B) ANOCVA with repeated measurements and duration of disease as covariate was performed for the times in the TWT. The between-subject factor is group, and the within-subject factor is condition (TWT-M, TWT-A, TWT-B). Group differences within a condition (e.g., TWT-M) were calculated with ANOVA. For the calculation of the dual-task costs (DTC), a 3 (group: Control, TD, PIGD) × 3 (condition: TWT-M, TWT-A, TWT-B) × 2 (interferences: motor vs. cognitive) ANOVA with repeated measurements was performed for the TWT. With significant results, post-hoc tests (Bonferroni correction) were used to check which factor levels significantly differ. An alpha value of 0.05 was used for all statistical tests (also for post-hoc analyses; [51]). In addition to the significance value (*p* < 0.05, * significant; *p* < 0.01; ** highly significant; *p* < 0.001, *** highly significant), the effect sizes for all ANOVAs were indicated using the partial eta^2^.

The times in the conditions of the TWT were measured using a stopwatch and expressed as 0.01 s. For the times in the TWT conditions (TWT-M; TWT-A, and TWT-B), the mean values (X) of the three runs were used:(1)$$\bar{X}\;=\frac{1}{n}\overset{1}{\underset{i=1}{\sum }}\mathrm{Xi}=\frac{X1+X2+X3}{n}$$

To calculate the DTCs, the performance under DT condition is related to the performance under ST condition. Since higher values indicate worse performance (times in the TWT), a negative sign was inserted. Negative DTCs indicate a deterioration compared to the ST condition [52]. Therefore, the motor and cognitive DTCs for the TWT are calculated as follows:(2)$$\mathrm{DTC}=\frac{\mathrm{Performance}\;\mathrm{in}\;\mathrm{DT}-\mathrm{Performance}\;\mathrm{in}\;\mathrm{ST}}{\mathrm{Performance}\;\mathrm{in}\;\mathrm{ST}}\times 100$$

TWT
(3)$$\mathrm{motorDTC}\;\mathrm{for}\;\mathrm{TWTA}\;\left(\%\right)=\frac{-\left(\mathrm{TWTA}-\mathrm{TWTM}\right)}{\mathrm{TWTM}}\times 100$$
(4)$$\mathrm{motorDTC}\;\mathrm{for}\;\mathrm{TWTB}\;\left(\%\right)=\frac{-\left(\mathrm{TWTB}-\mathrm{TWT}\right)}{\mathrm{TWTM}}\times 100$$
(5)$$\mathrm{cognitiveDTC}\;\mathrm{for}\;A\;\left(\%\right)=\frac{-(\mathrm{TWTA}-\left(\mathrm{TMTA}-\mathrm{TMTM}\right)}{\left(\mathrm{TMTA}-\mathrm{TMTM}\right)}\times 100$$
(6)$$\mathrm{cognitiveDTC}\;\mathrm{for}\;B\;\left(\%\right)=\frac{-(\mathrm{TWTB}-\left(\mathrm{TMTB}-\mathrm{TMTM}\right)}{\left(\mathrm{TMTB}-\mathrm{TMTM}\right)}\times 100$$

The Trail-Making Test [42] was used to evaluate cognitive ST performance. Due to the different lengths of the conditions in the TMT (TMT-M: 185.4 cm; TMT-A: 185.4 cm and TMT-B: 243.8 cm) [53], the velocities in all conditions are first calculated as follows:(7)$$\mathrm{Velocity}\;\mathrm{TMT}\;\mathrm{condition}\;\left(\frac{\mathrm{cm}}{s}\right)=\frac{\mathrm{Length}\;\mathrm{of}\;\mathrm{the}\;\mathrm{condition}}{\mathrm{Time}\;\mathrm{for}\;\mathrm{TMT}\;\mathrm{condition}}$$

The velocity was normalized to the length of 200 cm (required time for 200 cm):(8)$$\mathrm{Time}\;\mathrm{for}\;\mathrm{TMT}\;\mathrm{condition}\;\left(s\right)=\frac{200}{\mathrm{Velocity}\;\mathrm{TMT}\;\mathrm{condition}}$$

In addition, a “two-way” intra-class correlation coefficient (ICC) was calculated to quantify the consistency within the three trials of each TWT condition and the groups [54]. The test-retest reliability was assessed using the Standard Error of the Measurement [SEM = (SD × √(1 − ICC))] and the Minimum Detectable Change with a confidence interval of 95% [MDC95 = (1.96 × SEM × √(2))] [55]. In order to be able to compare both measures, these were additionally expressed as percentages (SEM% and MDC95%; [56]).

ROC (Receiver Operating Characteristic) analyses were performed to determine the diagnostic quality of the TWT, where sensitivity, specificity, and area under the curve (AUC) were considered (for the interpretation of the values, see [57]). The Youden index was used to determine which threshold was best suited to differentiate the groups [58].

## 3. Results

### 3.1. Characteristics of the Study Population

Sixteen PD patients with dominant tremor (TD, M = 68.2 years, SD = 8.94), 27 PD patients with postural instability and gait difficulty (PIGD, M = 69.7 years, SD = 8.68) and 20 healthy older adults (control, M = 72.4 years, SD = 5.53) participated in this study. In the MoCA, no differences between the groups can be observed. Compared to TD and the control group, an increased frequency of falls can be observed in persons with PIGD. Accordingly, fall-associated self-efficacy is significantly lower in PIGD. The proportion of mild to moderate depression is also higher in PIGD than in both other groups (see Table 3).

### 3.2. Reliability of Measurement Repetition in the TWT

Table 4 shows the relative and absolute reliability measures (ICC, SEM, MDC95). The inter-trial reliability is medium to excellent for all conditions and groups, with ICC values between 0.87 and 0.98. The reliability of the trials is between 0.87 and 0.98. In total, the SEM is between 0.22–3.20 s. The SEM% is low in all conditions and groups (0.51–4.05%). In 100% of the observations, a SEM% ≤ 10% can be found. The SEM varies between 0.26–2.68 s for the control group, 0.22–3.20 s for TD, and 0.43–2.18 s for PIGD. In total, the MDC95 is between 0.62–88.8 s for the absolute times in the TWT. The MDC95% fluctuated between 1.41–11.5% for the whole sample and is thus below ≤30%.

### 3.3. Times as Performance Measure in the TWT

The times in TWT-M and TWT-A are normally distributed in all groups (*p* < 0.05). The times in the TWT-B tend to be normally distributed (*p* = 0.069). Age (*r* = 215, *p* = 0.043) correlates significantly with the times in TWT-B. Sex tends to have a significant influence on performance in TWT-B (*p* = 0.093), with higher times observed for women (women: M = 79.4, SE = 3.74; men: M = 68.7, SE = 5.02). Education does not correlate with the performance in the conditions of the TWT (*r* = −0.064–0.035). Duration of disease, however, correlates with performance on TWT-A (*r* = 0.221, *p* = 0.04) and TWT-B (*r*= 0.249, *p* = 0.03), but not with performance on TWT-M (*r* = 0.128, *p* = 0.16). Across groups, we observe 2 outliers. One participant in the PIGD group with a value of M = 136.3 in the TWTB and one participant in the TWT-M with a value of M = 127.7, also in the PIGD group.

A 3 (condition: TWT-M, TWT-A, TWT-B) × 3 (group: Control, TD, PIGD) ANCOVA with repeated measurement of times for the TWT and duration of disease as covariate shows significant main effect for condition, *F*(1.24, 82.3) = 42.3, *p* < 0.001, *ɳ*^2^_p_ = 0.426, and group, *F*(2, 657) = 5.55, *p* < 0.05, *ɳ*^2^_p_ = 0.138. The post-hoc analysis shows that times are significantly higher in TWT-B (M = 74.1, SE = 3.01) than in TWT-A (M = 55.9, SE = 1.50; *p* < 0.001, 95%CI = 13.1–23.3) or in the purely motor condition (TWT-M: M = 45.1, SE = 1.88; *p* < 0.001; 95%CI = 22.6–35.5) for all subjects. PIGD (M = 65.9, SE = 2.58) differ from TD (M = 56.2, SE = 3.74; *p* = 0.108, 95%CI = −1.44–20.9) and significantly differ from the control group (M = 52.9, SE = 3.81; *p* < 0.05, 95%CI = 0.735–25.1). TD patients are not significantly different from the control group (*p* = 0.630, 95%CI = −16.3–5.24). A significant interaction of condition x group does not exist, *F*(2.88, 82.3) = 0.254, *p* = 0.851, *ɳ*^2^_p_ = 0.009). This shows that all groups walk slower with increasing cognitive load and therefore need longer (see Figure 1). A difference in the times in the TWT can thus be observed, in particular between PIGD and the control group. The covariate duration of disease has no significant effect, *F*(1, 57) = 0.582, *p* = 0.449, *ɳ*^2^_p_ = 0.010.

### 3.4. Motor-Related Cognitive Costs and Cognitive-Related Motor Costs in the TWT

Table 5 summarizes the mean values and standard deviations of the calculated DTC for the TWT.

Regarding the proportional DTC a 3 (group: Control, TD, PIGD) × 2 (condition: TWT-A, TWT-B) × 2 (interference: cognitive, motor domain) ANOVA with repeated measurement for the times in TWT was calculated. The results show significant major effects for condition, *F*(1, 60) = 19.5, *p* < 0.001, *ɳ*^2^_p_ = 0.245, and interference, *F*(1, 60) = 44.6, *p* < 0.001, *ɳ*^2^_p_ = 0.426. A significant interaction effect can be observed for condition x interference, *F*(1, 60) = 32.9, *p* < 0.001, *ɳ*^2^_p_ = 0.354. Post-hoc analysis shows that with low cognitive load, (TWT-A: M = −211, SE = 28.7) the performance losses are greater than they are with high cognitive load (TWT-B: M = −80.3, SE = 5.52; *p* < 0.001; 95%CI = −189.9–71.5).

Figure 2a shows the distribution of cognitive and motor interference in TWT-A in individuals with PD (PIGD & TD) and healthy controls. Most participants show mutual interferences with performance losses, especially in the cognitive task. Interferences in the motor task are low across all groups. The level of motor and cognitive interference and the range is comparable in all groups. Some participants show minor interference in the motor task but improvements in the cognitive task performance. However, group-specific patterns cannot be identified. In the condition with a high additional cognitive load (TWT-B, Figure 2b), the cognitive interferences are lower than in the TWT-A. Mutual interference can also be observed in TWT-B across groups. A few participants show low or positive interferences in the cognitive but deterioration in the motor task performance (cognitive-motor interference or cognitive task prioritization). However, clear group-specific patterns also cannot be identified.

Based on the calculated velocities, the TWT conditions allow for an appropriate differentiation between the motor phenotype PIGD and the control group (AUC > 0.8; see Table 5; grey marked cells). The TWT-A allows for a good differentiation (AUC = 0.831; sensitivity = 0.852; specificity = 0.800). However, the differentiation between phenotypes, PIGD and TD, is not satisfactory by any TWT conditions (AUC < 0.7). Only the TWT-M condition shows a significant result here as evidence of the accuracy of the test procedure (see Table 6; value in bold). The TWT-A tends to be significant (see Table 6; value in bold). Additionally, a distinction between TD and the control group is not sufficiently precise with any of the TWT conditions (AUC < 0.7).

On the other hand, the motor and cognitive DTC do not allow the groups to be differentiated. Sensitivity and specificity are insufficient to distinguish the groups from each other. As a result, the areas under the curve of the ROC analyses as a measure of accuracy are too small.

## 4. Discussion

This study aimed to evaluate the TWT as a potential method for quantifying postural instability and gait disturbances in PD patients and distinguishing between PD motor phenotypes. As expected, all participants in the study were slower under DT conditions [59]. The effect was greater in PIGD patients than in the control and TD groups. The greater the cognitive load, the greater the influence on walking performance. However, the difference between the groups became smaller with increasing cognitive load. The largest differences between the groups were found in the TWT-M (purely motor condition).

The TWT performance differs both overall and within the three groups as expected. Times increase with increasing cognitive load and is in line with the studies by Spildooren et al. [60], Wild et al. [61], and Kelly et al. [62]. They demonstrated increased difficulties and balance problems with locomotion tasks under DT conditions in PD (cf. the meta-analysis by Raffegeau et al. [15]). In particular, walking speed is significantly influenced in these studies. A significant difference between the two PD phenotypes can only be observed in the TWT-M and the task with low cognitive load (TWT-A). A distinction between PIGD and the control group becomes significant in all conditions. However, a 95% CI between 0.735—25.1 s shows that the confidence interval is very large. The values are very heterogeneous. The increased time by 25.1 s would be clinically relevant, but an increased processing time by 0.735 s is clinically less relevant. The difference between TD and the control group does not become significant in any condition. This can be explained by the fact that the TWT primarily claims aspects of mobility [27]. In the condition of high cognitive load, the control group also shows problems with the automatic execution of walking and increased walking times, so the group differences become smaller. Based on the calculated AUC values, a good discriminatory power is demonstrated to distinguish PIGD from individuals without PD (control group), with only TWT-A showing sufficient sensitivity (85.2%) and specificity (80%). Only moderate to poor AUC values can be found to differentiate between the two PD groups. These results suggest that one of the underlying mechanisms for gait dysfunction is cognition and slowed walking in complex situations may result from an overloaded or inefficient cognitive system in PIGD.

Regarding the motor and cognitive DTC, differences between the conditions of the TWT (TWT-A & TWT-B) can be observed. Higher motor DTCs (−66.3%) can be observed in condition TWT-B compared to −28.7% for TWT-A (*p* < 0.001). The magnitude of the motor DTC is larger than the studies summarized in the overview article by Kelly et al. [9]. In the studies by Kelly and colleagues, a range between −1% to −59% motor DTC is reported. This indicates that the TWT is significantly more demanding and requires more cognitive resources than walking straight ahead [63,64] or walking with a 180-degree turn [60,65]. In contrast, higher cognitive DTC can be observed in the TWT-A condition, with −392% compared to −94.1% for TWT-B (*p* < 0.001). In comparison to the few studies that also calculate DTC for the cognitive task (Galletly & Brauer, [66], with +31% and +72%, points to an improvement and prioritization of the cognitive task; O’Shea, Morris & Iansek, [67], with −5%; Yogev et al. [14], with −42%), in the present study cognitive DTCs can also be found, which are many times larger. With −392% (TWT-A) and −28.7% (TWT-B) cognitive DTC, high cognitive performance decrements can be observed, especially under low cognitive load. Moreover, there is a difference between the conditions of TWT in motor and cognitive DTC. All groups in the condition with high cognitive load (TWT-B) show greater motor-related cognitive DTCs compared to cognitive-related motor DTCs (*p* < 0.001). In the condition with low cognitive load (TWT-A), larger motor-related cognitive DTCs can be observed across all groups (*p* < 0.001). One possible explanation for the large cognitive DTC in TWT-A is that the relatively simple counting in ascending order is possible despite resource allocation toward the motor task, and the task can still be accomplished. In the TWT-B, on the other hand, the cognitive task (numbers and letters running alternately and in ascending order) requires more cognitive resources, which means that the limited attention resources [68] must be shifted in the direction of the cognitive task in order to complete TWT-B. Thus, a strategic allocation of attention resources is necessary to complete the TWT as successfully as possible. Contrary to theoretical expectations, no group-specific patterns of this allocation can be observed in this study. PD patients and the control group appear to have similar motor-cognitive interference patterns in complex locomotor tasks. Both PD groups and the control group show a risky allocation of resources towards cognitive tasks (“posture second” strategy) in DTs with high cognitive load (TWT-B) [36]. In the DT with lower cognitive load (TWT-A), on the other hand, an allocation of resources towards motor tasks with high cognitive DTC can be observed (“posture first” strategy) (see [61]).

There are some limitations in this study that need to be mentioned. Our sample size is very small. It is difficult to draw meaningful conclusions from analyses based on groups of 16, 27, and 20 participants. The study is cross-sectional. With longitudinal studies, changes over time can be mapped, and the prognosis can be improved. An explanation of the insufficient differentiation between the groups is that the classification into the mentioned motor phenotypes by the classification algorithm according to Jankovic et al. [33] only reflects the relationship between the cardinal symptoms (tremor and postural instability). For example, TD with strong tremors also showed significant constraints in balance control [69]. If the UPDRS is used to classify motor phenotypes, the scale (especially the UPDRS III; motor analysis) is fundamental. While some neurologists and researchers advocate the scale, others consider the scale to be a less representative snapshot of the current physical condition of PD patients. The test is based on subjective assessment by a neurologist and is highly dependent on the examiner’s expertise. In addition, the DTC was calculated based on the required times. Other measures are probably needed to show differences between the two PD motor phenotypes [70]. Gait parameters and their changes under DTs could allow for a more differentiated conclusion of the motor differences between these phenotypes [28,71] and improve the prognosis in the progression of PIGD.

## 5. Conclusions

Currently, there is no gold standard for assessing postural instability and gait disorders that encompasses all aspects of cognitive and physical characteristics in PD [71]. DTs, compared to STs, are often more sensitive in assessing these gait disorders [31,72]. Problems that remain undetected in the single-task condition can emerge with the use of DT paradigms.

PD therapy is primarily based on early detection and treatment of symptoms. The aim is to maintain the independence of those affected by the disease as long as possible to preserve the quality of life. Thus, patients with gait disorders must be identified by valid, well-evaluated clinical tests that precisely assess the individual fall risk and severity of the disease. Unfortunately, there is currently no gold standard for assessing postural instability and gait disorders that address all aspects of PD’s cognitive and physical characteristics [71]. The TWT covers various aspects of mobility and examines the relationship between cognitive functions and walking [27]. Especially the pure motor condition shows high ICC values and a SEM% below 1. Based on the results and concerning the sensitivity and specificity of the procedure, a differentiation into PD motor subtypes can be made as expected, especially with the purely motor condition of the TWT (TWT-M) and based on times. In future studies, it would be interesting to examine whether walking with directional changes (TWT) and an additional motor task (e.g., box-checking task; analogous to the studies by Heinzel, Maechtel, Hasmann, Hobert, Heger, Berg & Maetzler, [73]) generates more apparent differences in the DTCs between PD and a control group.

## Figures and Tables

**Figure 1 brainsci-12-01217-f001:**
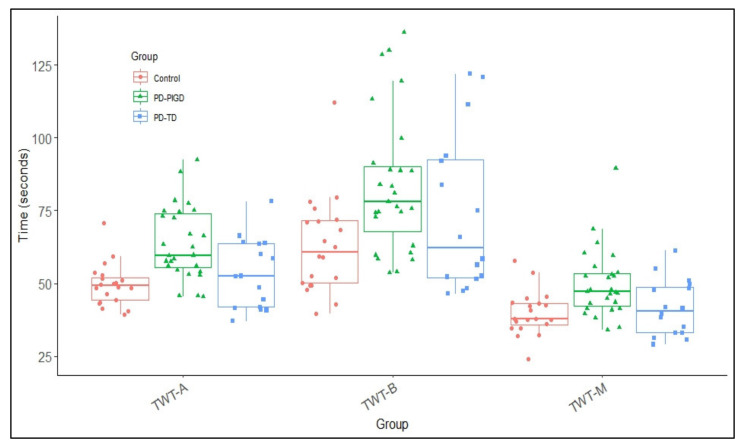
Mean and standard deviation of groups (PIGD, TD & control) and TWT conditions (TWT-A, TWT-B & TWT-M) based on times.

**Figure 2 brainsci-12-01217-f002:**
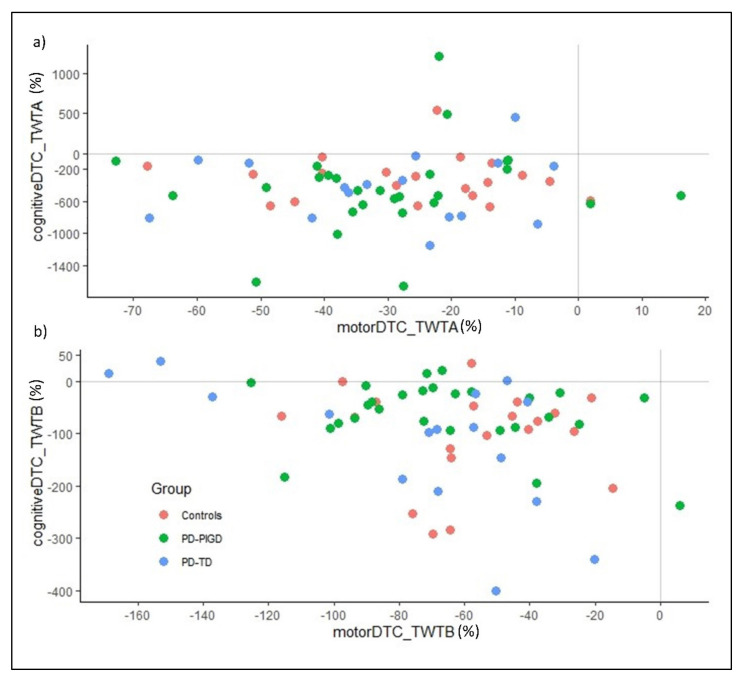
(**a**) Pattern of motor-cognitive DTC in TWT-A based on times in PIGD, TD, and control. (**b**) Pattern of motor-cognitive DTC in TWT-B based on times in PIGD, TD, and control. Motor and cognitive DTCs are calculated using the following formula: DTC (%) = ((performance DT − performance in ST)/performance in ST) ×100).

**Table 1 brainsci-12-01217-t001:** Inclusion and exclusion criteria for the control group, PD patients with TD, and PD patients with PIGD.

	Control	TD	PIGD
Inclusion criteria	- Either gender - Age 50 to 80 years of age - Ability to walk for 10 min continuously unassisted - Able to understand English instructions - Normal or corrected vision	- Either gender - Age 50 to 80 years of age - Diagnosed with idiopathic PD by a Neurologist - Ratio of mean tremor score/mean PIGD score (Jankovic-based classification) is 1.5 or more (with the use of UPDRS-II) - Ability to walk for 10 min continuously unassisted - Able to understand English instructions - Normal or corrected vision	- Either gender - Age 50 to 80 years of age - Diagnosed with idiopathic PD by a Neurologist - Ratio of mean tremor score/mean PIGD score (Jankovic-based classification) was less than or equal to 1.0 (with the use of UPDRS-II) - Ability to walk for 10 min continuously unassisted - Able to understand English instructions - Normal or corrected vision
Exclusion criteria	- A neurological disease other than PD - Had brain surgery in the past including implanted deep-brain-stimulation - Have significant co-morbidities likely to affect gait, e.g., history of stroke, - Peripheral neuropathy - Visual impairments that cannot be corrected - Clinically diagnosed with dementia (as stated in the patient’s information chart from the patient database at the Sun Life Financial Movement Disorders Research and Rehabilitation) - Are not able to comply with the protocol.	- Tremor score <4 or PIGD score >3 (with the use of UPDRS-II) - A neurological disease other than PD - Had brain surgery in the past including implanted deep-brain-stimulation - Have significant co-morbidities likely to affect gait, e.g., history of stroke, - Peripheral neuropathy - Visual impairments that cannot be corrected - Are unable to walk in the OFF state - Clinically diagnosed with dementia (as stated in the patient’s information chart from the patient database at the Sun Life Financial Movement Disorders Research and Rehabilitation) - Are not able to comply with the protocol.	- Tremor score >3 or PIGD score <4 (with the use of UPDRS-II) - A neurological disease other than PD - Had brain surgery in the past including implanted deep-brain-stimulation - Have significant co-morbidities likely to affect gait, e.g., history of stroke, - Peripheral neuropathy - Visual impairments that cannot be corrected - Are unable to walk in the OFF state - Clinically diagnosed with dementia (as stated in the patient’s information chart from the patient database at the Sun Life Financial Movement Disorders Research and Rehabilitation) - Are not able to comply with the protocol.

**Table 2 brainsci-12-01217-t002:** Items of the UPDRS to classify the motor subtypes Tremor Dominant (TD) and Postural Instability (PIGD) in PD patients.

UPDRS			
N Items	Tremor-Dominant (TD)	N Items	Postural Instability and Gait Difficulty (PIGD)
Part 2—Activities of daily living (ADL)			
1	2.16 Tremor	3	2.13 Falling (independent of rigidity) 2.14 Freezing during walking 2.15 Walking
**Part 3—Motor examination**			
7	3.20 Rest Tremor, F 3.20 Rest Tremor, RH 3.20 Rest Tremor, LH 3.20 Rest Tremor, RF 3.20 Rest Tremor, LF 3.21 Action or posture tremor of the hands, L 3.21 Action or posture tremor of the hands, R	2	3.29 Gait 3.30 Postural Stability *

Note. F = face; RH = right hand; LH = left hand; RF = right foot; LF = left foot; R = right; L = left; * Reaction to sudden rearward displacement by pulling on the patient’s shoulders; standing straight with eyes open and feet slightly apart (the patient is prepared). The ratio of the mean TD scores (8 items) to the mean PIGD scores (5 items) was used to classify the motor subtypes: TD (ratio ≤ 1.5), PIGD (ratio ≥ 1).

**Table 3 brainsci-12-01217-t003:** Characteristics of PD patients differentiated into motor phenotypes TD and PIGD, including mean values (standard deviation) and test values of UPDRS-III.

	Control	TD Ratio ≤ 1.5	PIGD Ratio ≥ 1.0	Stat. Analyses
	(*n* = 20)	(*n* = 16)	(*n* = 27)	
sex	7 men, 13 women	11 men, 5 women	23 men, 4 women	CHI2(2) = 14.2 **
age (years)	72.4 (5.53)	68.2 (8.94)	69.7 (8.68)	*F*(2, 60) = 1.32, *ɳ*^2^_p_ = 0.042
BMI (kg/m^2^) Under-, Normal-, Obesity (*n*)	27.9 (4.73); 0, 6, 6, 5	24.2 (4.40) ^ŧ^; 2, 6, 6, 1	27.4 (4.07); 0, 8, 12, 5	*F*(2, 57) = 3.79 *, *ɳ*^2^_p_ = 0.117
UPDRS-III (Score; max = 108)	-	22.53 (7.47)	23.7 (7.97)	t(41) = 0.646, d = 0.147
Duration of the disease (years)	-	6.19 (4.92)	4.93 (4.37)	t(41) = 0.388, d = 0.275
Activities Specific Balance Confidence (ABC)Scale, %	95.3 (3.77)	89.7 (8.24)	79.3 (19.9) ^ŧ^	*F*(2, 59) = 8.05 **, *ɳ*^2^_p_ = 0.214
Fall experience last year (n persons; n in %, *n* falls)	3 persons (15%); 3 falls	4 persons (26.7%); 6 falls	9 persons (33%); 31 falls	*F*(2, 14) = 1.68, *ɳ*^2^_p_ = 0.222
Timed Up-and-Go test (TUG), seconds	8.55 (1.29)	9.19 (2.28)	11.1 (2.99) ^ŧ^	*F*(2, 61) = 7.29 *, *ɳ*^2^_p_ = 0.198
Montreal Cognitive Assessment (MoCA), score	27.9 (1.48)	26.9 (3.23)	27.6 (1.95)	*F*(2, 60) = 0.854, *ɳ*^2^_p_ = 0.028
	2 participants with a score below 26	4 participants with a score below 26	6 participants with a score below 26	*CHI^2^*(2) = 1.61
Education (years)	10.5 (0.76)	13.7 (4.17)	13.5 (3.40) ^ŧ^	*F*(2, 61) = 6.62 **, *ɳ*^2^_p_ = 0.189
Geriatric Depression Scale (GDS), *n*	20 normal 0 mild to moderate 0 severe	15 normal 1 mild to moderate 0 severe	22 normal 5 mild to moderate 0 severe	*CHI^2^*(2) = 3.21

Note. ** *p* < 0.01; * *p* < 0.05; ^ŧ^ significant difference to control group (*p* < 0.05).

**Table 4 brainsci-12-01217-t004:** Intra-class correlation (ICC) and absolute inter-trial reliability (SEM) across the three TWT conditions.

	Control			TD			PIGD		
	ICC (95% CI)	SEM/ SEM (%)	MDC95/MDC95%	ICC (95% CI)	SEM/ SEM (%)	MDC95/MDC95%	ICC (95% CI)	SEM/ SEM (%)	MDC95/MDC95%
TWT-M	0.974 (0.95–0.99)	0.26/ 0.68	0.76/ 1.89	0.987 (0.97–0.99)	0.22/ 0.51	0.62/ 1.41	0.959 (0.92–0.98)	0.43/ 0.74	1.18/ 2.05
TWT-A	0.894 (0.78–0.96)	1.15/ 2.31	3.19/ 6.39	0.959 (0.90–0.98)	0.72/ 1.33	1.99/ 3.68	0.939 (0.88–0.97)	1.03/ 1.63	2.86/ 4.54
TWT-B	0.870 (0.72–0.94)	2.68/ 4.05	7.43/ 11.21	0.886 (0.74–0.96)	3.20/ 4.14	8.88/ 11.48	0.918 (0.84–0.96)	2.18/ 2.59	6.02/ 7.17

Note. In order to calculate the reliability measures, 3 trials (measurement time points) were included; CI = confidence interval; SEM = standard error of measurement; MDC = minimal detectable change.

**Table 5 brainsci-12-01217-t005:** Mean values and standard deviation of DTC in the TWT divided into the PD phenotypes and healthy controls.

	PIGD (n = 27)	TD (n = 16)	Control (n = 20)	Statistical Analysis
Motor DTC TWT-A	−29.9 (18.4)	−29.7 (18.7)	−26.6 (17.6)	*F*(2, 60) = 0.220, *p* = 0.803, *ɳ*^2^_p_ = 0.007
Motor DTC TWT-B	−65.4 (31.9)	−75.4 (43.1)	−58.1 (26.8)	*F*(2, 33) = 1.16, *p* = 0.320, *ɳ*^2^_p_ = 0.037
Cognitive DTC TWT-A	−431 (543)	−430 (417)	−317 (284)	*F*(2, 33) = 0.445, *p* = 0.643, *ɳ*^2^_p_ = 0.015
Cognitive DTC TWT-B	−61.2 (62.2)	−118 (126)	−103 (90.8)	*F*(2, 33) = 2.33, *p* = 0.106, *ɳ*^2^_p_ = 0.072

Note. DTC = dual-task costs; the empirical mean values and standard deviations are shown; PIGD: Parkinson-Postural Instability and Gait Difficulty; TD: Parkinson-Tremor Dominant; Control: older adults without Parkinson diagnosis.

**Table 6 brainsci-12-01217-t006:** Statistics and receiver operating characteristic curve thresholds for the TWT (velocities in the TWT; motor DTC) to differentiate between PIGD, TD, and the control group.

Condition	Groups	n	Youden Index	Sensitivity	Specificity	Threshold	AUC	*p*
**TWT-M**	PIGD vs. TD	27/16	0.326	0.889	0.438	1.05	0.685	**0.044**
	PIGD vs. Control	27/20	0.530	0.630	0.900	0.891	0.791	**<0.001**
	TD vs. Control	16/20	0.288	0.436	0.850	0.914	0.553	0.588
**TWT-A**	PIGD vs TD	27/16	0.352	0.852	0.500	0.778	0.662	0.079
	PIGD vs. Control	27/20	0.652	0.852	0.800	0.776	0.831	**<0.001**
	TD vs. Control	16/20	0.400	0.500	0.900	0.711	0.638	0.161
**TWT-B**	PIGD vs. TD	27/16	0.303	0.741	0.563	0.593	0.623	0.183
	PIGD vs. Control	27/20	0.541	0.741	0.800	0.567	0.783	<0.001
	TD vs. Control	16/20	0.325	0.375	0.950	0.503	0.613	0.252
**Motor DTC TWT-A**	PIGD vs. TD	27/16	0.234	0.296	0.938	−16.52	0.588	0.340
	PIGD vs. Control	27/20	0.356	0.556	0.800	−37.81	0.659	0.064
	TD vs. Control	16/20	0.300	0.500	0.800	−38.13	0.597	0.324
**Motor DTC TWT-B**	PIGD vs. TD	27/16	−0.093	0.407	0.500	−241.1	0.479	0.821
	PIGD vs. Control	27/20	−0.089	0.111	0.800	−133.7	0.513	0.880
	TD vs. Control	16/20	0.188	0.938	0.250	−378.7	0.541	0.679
**Cognitive DTC TWT-A**	PIGD vs. TD	27/16	0.264	0.889	0.625	−823.6	0.528	0.763
	PIGD vs. Control	27/20	−0.219	0.481	0.300	−493.3	0.431	0.426
	TD vs. Control	16/20	−0.375	0.625	0.000	−786.5	0.413	0.373
**Cognitive DTC TWT-B**	PIGD vs. TD	27/16	0.215	0.778	0.438	−35.53	0.567	0.466
	PIGD vs. Control	27/20	0.344	0.444	0.900	2.65	0.615	0.182
	TD vs. Control	16/20	0.225	0.375	0.850	−3.94	0.544	0.656

Note. n = number of cases; *p* = significance value; PD = Parsinson Disease; PIGD = Postural Instability/Gait Difficulty; TD = Tremor Dominant; DTC = dual-task costs; TWT = Trail-Walking Test; AUC (AUROC) = Area Under the Receiver Operating Characteristic Curve; For continuous variables, limit values were determined from the optimal combination of sensitivity and specificity using the Youden index; the relevant data mentioned in the text are highlighted in the table by the grey cells and significant results are highlighted in bold.

## Data Availability

All relevant data are within the study, and raw data are available on request.

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
