# Peer review of "Profiles of Motor-Cognitive Interference in Parkinson’s Disease—The Trail-Walking-Test to Discriminate between Motor Phenotypes"

_brainsci, 2022, doi:10.3390/brainsci12091217_

Round 1

Reviewer 1 Report

Figure 1. I think it would be helpful to denote the time measure (seconds ?) on Y-axis.

Figures 2-3. Similarly, what is the measure of DTEc (% ?) In addition, I did not find a description for DTEc and DTEm neither in legends to these figures nor in the text. I have a guess what do they mean, but still it must clearly wtitten in the methods section.

There is abundance of acronyms across the text, what slightly hinders reading. Consider, for example, to use just TD and PIGD instead of PD-TD and PD-PIGD.

Author Response

Dear Reviewer,

Thank you for your comments. We have addressed all your comments and think that the quality of our article improved accordingly.

You can see our changes in the Word file.

Kind regards

Reviewer 2 Report

The authors investigated if the Trail-Walking Test is a potential diagnostic instrument to improve the predictive power of the clinical evaluation concerning the severity of the disease and record the different aspects of walking. They included 20 controls, and 43 PD patients assessed with TWT and other neuropsych tests, demographic characteristics, PD-related data. Results suggest that a differentiation into PD motor subtypes can be made, especially with the purely motor condition of the TWT (TWT-M) and based on times.

The manuscript is well written, structured and the methods are appropriate. The study itself is ambitious and the data is interesting. Nonetheless, I would like to discuss some issues that may need to be addressed in the manuscript. They are listed below not necessarily in order of importance.

The introduction in general is very good but perhaps at the beginning, when talking about dementia, it would be good to include the following reference:

Hely M. A., Reid W. G., Adena M. A., Halliday G. M., Morris J. G. (2008). The Sydney multicenter study of Parkinson’s disease: the inevitability of dementia at 20 years. Mov. Disord. 23, 837–844 10.1002/mds.21956

There is a need to clarify here the objectives or main goals of this study to facilitate the reading.

Did the authors assessed the normality of the variables at baseline? Is all the data parametric? 

I would recommend examine the effect size (Cohen d and 95% confidence interval [CI]) and standard error based on change score differences between groups to investigate the clinically relevance of the results and explain it in the discussion.

The Discussion section would probably benefit if the authors provided more information about the implications of their findings (instead of merely comparing their results with the results of other studies). The authors should also emphasize what the readers can learn from their study.

In the abstract and methods 46 PD patients are noted but in tables only 43. Is this difference due to exclusion criteria? 

Please explain in more detail exclusion/inclusion criteria and if there is financial compensation for their participation.

What participants are the outliers in the PD groups? In the figure 1 some of them could be potential outliers. In order to understand potential bias in sample selection, could the authors clarify this point. Also, if the participants are excluded of the analysis, are the excluded patients different in somehow from the included PD patients?

Did the authors examine if age of disease onset was of importance?

Apart from neuropsychological measures, any functional or emotional/quality of life test was used? Also it would be interesting to evaluate activities of daily living and how these measures can influence directly the neuropsychological outcomes. 

The authors assess global cognition with a screening test. It would be interesting to know what percentage of the sample has mild cognitive impairment. What percentage of the sample has MCI? Do the results maintain if we exclude these participants? And if we perform analysis with subgroups based on MCI?

To describe the sample, the authors divide it into men and women. It would be interesting to investigate whether the results are the same as a function of sex. In fact, in the control group, there are more women than men. Has it been taken into account or has it been analyzed?

Demographical information is missed. It would also be interesting to see if there are differences in years of education or cognitive reserve in these participants. I think a table with demographics of the sample would be desirable. It would be interesting to control education as covariate.

On what dates was the recruitment made? Has there been recruitment before and after the pandemic?

Sample size is very small. It is difficult to draw strong conclusions from analyses based on group of 16, 27 and 20 participants.

Did any of the PD subjects suffer from treatment related complications?

Has Hoehn & Yahr scale been used? I recommend to include it in the demographic table.

Minor comments:

TWT in the abstract

Table 5 significant result in bold

Line 218 is 3 condition instead of 2?

Author Response

Dear Reviewer,

thank you for your comments. We have addressed all your comments and think that the quality of our article improved accordingly.

You can see our changes in the Word file.

Kind regards

Round 2

Reviewer 2 Report

The authors have addressed all comments and I believe the article can be published. Thank you for responding to the comments and for reviewing the issues I suggested.